# Tumor Immune Microenvironment Clusters in Localized Prostate Adenocarcinoma: Prognostic Impact of Macrophage Enriched/Plasma Cell Non-Enriched Subtypes

**DOI:** 10.3390/jcm9061973

**Published:** 2020-06-24

**Authors:** Neil K. Jairath, Mark W. Farha, Sudharsan Srinivasan, Ruple Jairath, Michael D. Green, Robert T. Dess, William C. Jackson, Adam B. Weiner, Edward M. Schaeffer, Shuang G. Zhao, Felix Y. Feng, Issam El Naqa, Daniel E. Spratt

**Affiliations:** 1Department of Radiation Oncology, University of Michigan, Ann Arbor, MI 48104, USA; jairathn@med.umich.edu (N.K.J.); markfarh@med.umich.edu (M.W.F.); sudhu@med.umich.edu (S.S.); rj79@evansville.edu (R.J.); migr@med.umich.edu (M.D.G.); rdess@med.umich.edu (R.T.D.); wcj@med.umich.edu (W.C.J.); shgzhao@med.umich.edu (S.G.Z.); ielnaqa@med.umich.edu (I.E.N.); 2Department of Urology, Northwestern University, Chicago, IL 60611, USA; adam.weiner@northwestern.edu (A.B.W.); edward.schaeffer@northwestern.edu (E.M.S.); 3Department of Radiation Oncology, UCSF, San Francisco, CA 94143, USA; felix.feng@ucsf.edu; 4Veterans Affair Ann Arbor Healthcare System, University of Michigan, Ann Arbor, MI 48104, USA

**Keywords:** prostate cancer, immunotherapy, radiation therapy, macrophages, RNAseq, tumor immune microenvironment, plasma cells

## Abstract

Background: Prostate cancer (PCa) is characterized by significant heterogeneity in its molecular, genomic, and immunologic characteristics. Methods: Whole transcriptome RNAseq data from The Cancer Genome Atlas of prostate adenocarcinomas (*n* = 492) was utilized. The immune microenvironment was characterized using the CIBERSORTX tool to identify immune cell type composition. Unsupervised hierarchical clustering was performed based on immune cell type content. Analyses of progression-free survival (PFS), distant metastases, and overall survival (OS) were performed using Kaplan–Meier estimates and Cox regression multivariable analyses. Results: Four immune clusters were identified, largely defined by plasma cell, CD4^+^ Memory Resting T Cells (CD4 MR), and M0 and M2 macrophage content (CD4 MR^High^Plasma Cell^High^M0^Low^M2^Mid^, CD4 MR^Low^Plasma Cell^High^M0^Low^M2^Low^, CD4 MR^High^Plasma Cell^Low^M0^High^M2^Low^, and CD4 MR^High^Plasma Cell^Low^M0^Low^M2^High^). The two macrophage-enriched/plasma cell non-enriched clusters (3 and 4) demonstrated worse PFS (HR 2.24, 95% CI 1.46–3.45, *p* = 0.0002) than the clusters 1 and 2. No metastatic events occurred in the plasma cell enriched, non-macrophage-enriched clusters. Comparing clusters 3 vs. 4, in patients treated by surgery alone, cluster 3 had zero progression events (*p* < 0.0001). However, cluster 3 patients had worse outcomes after post-operative radiotherapy (*p* = 0.018). Conclusion: Distinct tumor immune clusters with a macrophage-enriched, plasma cell non-enriched phenotype and reduced plasma cell enrichment independently characterize an aggressive phenotype in localized prostate cancer that may differentially respond to treatment.

## 1. Introduction

Globally, there were 358,989 deaths from prostate cancer (PCa) in 2018, and there are expected to be 378,553 deaths from PCa in 2020 [1,2]. Despite the largely favorable survival rates for non-metastatic patients, the disease is characterized by an important heterogeneity from molecular, genomic and ultimately clinical standpoints [3,4,5]. Historically, standard clinicopathological factors including prostatic specific antigen (PSA), T category, Gleason score, and various combinations of these factors have been used for risk stratification and management decisions. Yet, increasing evidence has shown that these parameters are not accurate prognosticators for patients with newly diagnosed PCa [6,7]. There remains a need to develop more effective risk-stratification tools and predictive biomarkers to guide treatment. The advent of rapid sequencing technologies has opened the door across oncology to more precise targeting of treatment modalities tailored to specific patient subgroups.

The tumor immune microenvironment (TIME) has been shown to have prognostic significance and implications for therapeutic response in many different cancers [8,9,10]. The heterogeneity of tumors’ molecular and cellular composition is exacerbated during the progression of the cancer, creating complexity in the cellular and noncellular components of the tumor niche—the tumor microenvironment (TME). This complexity may relate to spatial and temporal phenomena, polarization of cellular phenotypes due to plasticity, and genomic or epigenomic instability [11,12]. Cell types across the immune system may be found within the TIME, but depending on varying molecular signals, the same immune cell type may promote or inhibit tumor progression [13,14]. This implies the multifactorial nature of the TIME, involving the interplay between antigen presentation, immune activation and immune suppression.

Cancer cells with inherent genetic instability form abnormal proteins that have not been previously recognized by the immune system, and these proteins become immunogenic antigens (neoantigens) that can trigger CD8+ T-cell responses via macrophage presentation or spontaneously. Cancer cells that present immunogenic neoantigens are eliminated from the host through the process of immune surveillance [15]. Cancers then promote an immune-suppressive TIME in which immune-suppressing cells, including tumor-associated macrophages (TAMs), regulatory T (Treg) cells, and MDSCs exist. Although prognostic implications of TIME composition have been demonstrated for breast, lung, liver, pancreatic, gastrointestinal cancers, and malignant melanoma [16], in PCa there is emerging data for the heterogeneous impact of specific cell types [17]. Given the complex interplay of the immunologic composition, we sought to characterize the presentation and clinical outcomes of discrete immune clusters in localized PCa, and to test whether these TIME subtypes inform prognosis.

## 2. Experimental Section

### 2.1. Study Design

We conducted a retrospective analysis of patients with histological diagnosis of prostate adenocarcinoma (PRAD) in the TCGA database. Samples came from institutions across Australia, Brazil, Canada, Georgia, Germany, Moldova, Romania, Russia, the United States, Vietnam, and Yemen. Each patient with clinical and genomic data was independently reviewed by two authors (N.K.J. and M.W.F.). Patients with metastatic disease at the time of diagnosis were not included in the study.

### 2.2. Clustering Based on Immune Cell Subpopulations

The CIBERSORT “in silico flow cytometry” tool was used to quantify the relative levels of distinct immune cell types in the TCGA PRAD dataset [18]. A mixture file containing the RNA Seq by Expression Maximization (RSEM) gene expression data from the samples in the TCGA PRAD dataset was downloaded from the cBioPortal and formatted according to the guidelines outlined in the CIBERSORT manual [19,20]. The LM22 signature genes file was used as a reference point for comparison. LM22 contains 547 genes that accurately distinguish 22 mature human hematopoietic populations and activation states, including seven T cell types, naïve and memory B cells, plasma cells, NK cells and myeloid subsets. The LM22 file was constructed from the gene expression profiles of those cell types measured on Affy U133A/Plus2 and Illumina Expression BeadChip (HumanHT-12 v4) platforms. By default, CIBERSORT estimates the relative fraction of each cell type in the sample, such that the sum of all relative fractions for each of the 22 cell subsets is equal to 1 for the sample. Designations of “High” and “Low” were created using median expression values based on the relative fractions of each cell type in the total cohort. CIBERSORT was also used to produce a quantitative score that measures the overall abundance of each cell type. The absolute immune fraction score was compared across cell types and samples.

The package “ComplexHeatmap” was downloaded from Bioconductor [21]. Unsupervised hierarchical clustering was performed by sample rather than by cell type. The clustering distance metric was set to maximum distance between rows, and the clustering method was the Ward’s minimum variance. The dataset was divided into four clusters based on visual interpretation of the immune cell subpopulation distribution, minimizing intracluster heterogeneity. Sample IDs corresponding to each cluster were extracted for further analysis.

### 2.3. Survival Analysis of Clusters

The primary outcome in our analysis was progression-free survival (PFS), defined as the time from pathologic diagnosis to the first occurrence of a new tumor, including biochemical recurrence (BCR), locoregional recurrence (LRR), distant metastases (DM), or death. Overall survival (OS) was also evaluated and defined as the time from pathologic diagnosis to death or loss to follow-up, as defined by the TCGA study group. Treatment details including timing and dose of radiation was obtained from the Firebrowse web tool [22]. Patients still alive were censored at the time of last follow-up. Each endpoint was assessed using the Kaplan–Meier method, and survival curves were compared using the Mantel–Cox log-rank test. Survival analysis was carried out using the survminer R package (R Foundation for Statistical Computing Version 3.6.2, Vienna, Austria) [23]. Log-rank *p*-value and risk tables are displayed on each chart.

### 2.4. Retrieving Raw Count Data

The “TCGAbiolinks” package was downloaded from Bioconductor [24,25,26]. The GDC query function was used to retrieve Illumina HiSeq RNA data from primary tumors in the TCGA PRAD dataset and the data were downloaded using the GDCdownload function. Raw count data were normalized, and low count genes were filtered according to the default 25% quantile across all samples. The table of normalized and filtered raw count data was extracted for use in downstream analysis.

### 2.5. Statistical Analysis

Clinical, pathologic, and molecular characteristics were compared between clusters using statistical methods in R using a chi-square test. The Kaplan–Meier method was used to compare endpoints across clusters with the log-rank test. To determine the influence of immunologic and clinicopathologic covariates on PFS, a multivariable Cox regression analysis was performed. Statistical significance was set at *p* < 0.05.

## 3. Results

### 3.1. TIME Clustering

Patients were divided into four distinct clusters based on the relative fractions of immune cell subsets within the TIME; 139 (28.0%) patients were grouped into cluster 1 (CD4 Memory Resting T cell [CD4 MR]^High^Plasma Cell^High^M0^Low^M2^Mid^), 115 (23.2%) in cluster 2 (CD4 MR^Low^Plasma Cell^High^M0^Low^M2^Low^), 112 (22.6%) in cluster 3 CD4 MR^High^Plasma Cell^Low^M0^High^M2^Low^), and 126 (25.4%) into cluster 4 (CD4 MR^High^Plasma Cell^Low^M0^Low^M2^High^) (Figure 1A). The absolute immune cell infiltration in each cluster was determined by the CIBERSORT absolute immune score output and displayed in Figure 1B. Pairwise t-tests were conducted to confirm significant differences between cell types between clusters (Table A1). All clusters demonstrated significantly different levels of each type of immune cell infiltrate (*p* < 0.05) except for M0 macrophages in clusters 1 versus 2 (*p* = 0.937) and CD4 MR in clusters 3 vs 4 (*p* = 0.46). Only immune cell subsets with non-zero median expression were displayed in Figure 1. Full unsupervised hierarchical clustering for 22 immune cell subsets can be visualized in Figure A1.

### 3.2. Baseline Demographic and Treatment Details by TIME Cluster

Complete demographic variables by cluster can be seen in Table 1. Patients in the macrophage enriched/plasma cell non-enriched clusters (3 and 4) had higher rates of Gleason 8–10 (25%, 29%, 52%, and 60% for clusters 1–4), high risk disease (30%, 34%, 46%, 48% for clusters 1–4), and older age (60, 61, 63, and 65 years old) compared to plasma cell high subtypes (clusters 1 and 2). TMPRSS2:ERG fusion status was similar across clusters, as was predominant tumor location within the prostate gland. Post-operative radiotherapy (adjuvant or salvage) was more common in the macrophage enriched/plasma cell non-enriched clusters (13%, 15%, 25%, 23% for clusters 1–4), as was salvage hormone therapy or chemotherapy (9%, 14%, 22%, 20% for clusters 1–4). Post-operative radiotherapy dose was similar across clusters (66–70 Gy).

### 3.3. TIME Cluster Prognostic Effect

The median follow-up time for all patients was 31 months and was similar across clusters (28, 34, 31, and 31 months for clusters 1-4, respectively). Overall, 9 patients (1.8%) had died at the time of this study due to their prostate cancer, and 82 (16.5%) had recurrence of their disease. Of those that recurred, 68 (13.7%) had a BCR, 7 (1.4%) had LRR, and 7 (1.4%) had DM.

The primary endpoint of the study, PFS, was analyzed for each cluster using Kaplan–Meier estimates and compared using the log-rank test. PFS, or median time from pathologic diagnosis to a new tumor event, defined as BCR, LRR, DM, or death was 29.5 months in cluster 1, 30.1 months in cluster 2, 23.1 months in cluster 3, and 26.1 months in cluster 4 (global *p*-value < 0.0001, Figure 2). The two clusters with the poorest performance in the PRAD cohort were clusters 3 and 4, as demonstrated by pairwise comparisons, both of which uniquely expressed a macrophage-enriched and plasma cell non-enriched phenotype in comparison to clusters 1 and 2.

When clusters 3 and 4 were pooled and compared to clusters 1 and 2, clusters 3 and 4 (macrophage enriched/plasma cell non-enriched) had shorter PFS (HR 2.24, 95% CI 1.46–3.45, *p* = 0.0002, Figure 3A), increased BCR (HR 2.13, 95% CI 1.30–3.48, *p* = 0.0026, Figure 3B), DM (all metastatic events occurred only in clusters 3 and 4, *p* < 0.0001, Figure 3C) and no significant difference in OS (HR 0.932, 95% CI 0.249–3.49, *p* = 0.917, Figure 3D). These results indicate a poorer PFS and OS in patients who have tumors with macrophage-enriched and plasma cell non-enriched phenotypes.

Subgroup analyses were then performed on clusters 3 and 4. When comparing PFS for only patients receiving surgery in cluster 3 vs. 4, cluster 4 displayed significantly worse PFS (HR 18.1, 95% CI 2.41–135.5, *p* < 0.0001, Figure 4A). This serves to demonstrate a poor prognosis correlated with M2 macrophages in PRAD. In contrast, when isolating only patients who received surgery and radiation, cluster 3 displayed a poorer prognosis, with median PFS of 29.2 months in cluster 4 and 20 months in cluster 3 (HR 0.37, 95% CI 0.157–0.864, *p* = 0.0217), Figure 4B). This indicates a correlation between patients with M0-enriched phenotypes and poorer PFS with radiation therapy. A test for interaction revealed no significant interaction for the outcome of PFS in the fitted Cox regression analysis between treatment groups and clusters (*p* = 0.5640).

After establishing that macrophage-enriched, plasma cell non-enriched PCa offers the poorest prognosis of the immune subpopulations, a multivariable analysis was performed to determine the independent prognostic impact of TIME clusters on PFS (Figure 5). Only patients with complete clinical data were included in the multivariable analysis. Cluster 3 (cluster 1 was reference, HR 4.27, 95% CI 2.08–8.77, *p* < 0.001) was independently prognostic for the endpoint of PFS. Cluster 4 trended towards also having independently worse prognosis (HR 1.67, 95% CI 0.87–3.19, *p* = 0.121), but did not reach statistical significance. Relative fraction of M0 macrophages in the tumor sample was independently prognostic for PFS (HR 1.42 per 0.1 unit increase, 95% CI 1.13–1.72, *p* = 0.019), while an increased relative fraction of plasma cells seemed to demonstrate a protective effect (HR 0.63 per 0.1 unit increase, 95% CI 0.51–0.77, *p* < 0.001). Tumor-associated macrophages, particularly M0 macrophages, and plasma cells appear to exert significant yet opposite effects on tumor progression and recurrence in PRAD, which converge to create the poor prognosis seen in cluster 3.

## 4. Discussion

In this retrospective analysis of whole-exome sequenced samples, we characterized the landscape of the TIME in localized PCa and identified four distinct immune clusters that are defined by four cell types. The two clusters (3 and 4) with low plasma cell content and expression of macrophages (M0 and M2, respectively) had worse outcomes compared to the plasma cell enriched clusters (1 and 2). M0 macrophages are plastic cells that can change their phenotype, potentially shifting toward pro-tumor TAMs, under the influence of environmental signals such as radiation damage [27]. In PCa, TAMs and other myeloid subsets are known to constitute up to 70% of tumor immune subsets, and they are known to influence tumor progression through angiogenesis, tumor cell proliferation, control over adaptive immunity, and metastasis formation [28,29,30,31,32,33,34]. Other studies have shown that TAMs confer poor prognosis in prostate cancer, likely through their ability to promote metastasis and progression through the CCL2-CCR2 and CCL22-CCR4 axes [17,35,36,37,38]. For this reason, TAMs have been an attractive target of therapy in cancer patients, and strategies to deplete these cell populations have been investigated in the clinical setting [31,39,40]. However, the reported overall benefit from these therapies has been negative to date [41]. This may be due to the extreme plasticity of these cells, as TAMs may be either pro-tumoral or anti-tumoral, and blunting the pro-tumoral effects may be offset by the depletion of anti-tumoral macrophage populations [41].

In this analysis, an abundance of plasma cells is significantly associated with better prognosis in the TCGA PRAD cohort. A meta-analysis of 21 studies of primary tumors concerning of the role of plasma cells in various cancers recently demonstrated the positive prognostic value of plasma cells in a number of cancer types, including breast, colorectal, lung, ovarian, gastric, hepatocellular, and pancreatic cancers, as well as melanoma [42]. However, this same analysis revealed that the histologic subtype of plasma cells is an important consideration in determining prognosis, a nuance that this analysis may not adequately capture. Another recent study recapitulated these findings, determining through CIBERSORT analysis that plasma cells are significant predictors of favorable survival across solid tumors in general [43]. Our analysis supports the findings of previous studies in classifying plasma cells as significant predictors of favorable prognosis in PRAD.

There is growing interest in immunoradiotherapy. A recent study found PD-L2, a checkpoint inhibitor targeting PD-1, to have significant interaction with predicted radiation therapy response [17]. Another recent study demonstrated that radiation therapy may lead to the polarization of macrophages toward the immunosuppressive, pro-tumoral phenotype [44]. In fact, both in vitro and in vivo experiments have demonstrated that TAMs from irradiated tumors support tumor growth and resistance to radiation therapy [45,46]. Thus, strategies for eliminating or inhibiting macrophages in conjunction with radiation therapy have demonstrated enhanced anti-tumor efficacy [47,48]. One animal study conducted in PCa with CSF1R signaling blockade, and one phase 2 clinical trial with sunitinib, did not show significantly improved results over radiation alone [49,50]. These studies did not account for the noncellular components of the TIME that may have influenced these results. To the authors’ knowledge, there are no therapies targeting plasma cells in prostate cancer. Our analysis demonstrated that within patients receiving post-operative radiotherapy, cluster 3, dominated by M0 macrophages and demonstrating a paucity of plasma cells, especially has suboptimal outcomes. As has been shown in prior literature, M2 macrophages are correlated with poor prognosis in our dataset, evidenced by the poor PFS of cluster 4 compared to clusters 1 and 2, and compared to cluster 3 in the surgery-only cohort. The M0 macrophage has long been thought to be a bystander in cancer, however we propose a potential role of these cells as harbingers of poor prognosis upon radiation exposure that requires validation. These results support the potential benefit for further research into immunoradiotherapy targeting macrophages and plasma cells in prostate cancer.

Our study is not without limitations. This is a retrospective analysis and is therefore subject to the limitations associated with this exploratory study design. Cell types were approximated through RNAseq algorithms, rather than direct assessment (i.e., immunohistochemistry). The median follow-up in TCGA for PCa is short and the number of events are limited. Thus, smaller effect sizes are likely to not be captured in our analyses.

Our study has built on the previous literature by characterizing immune clusters within localized PCa [17]. Additionally, by characterizing the aggressive nature of macrophage-enriched/plasma cell non-enriched PCa, particularly the prognostic impact of M0 (undifferentiated) macrophages, these results shed light on a potential risk stratification tool and rational subgroups in which to test TIME-based treatment strategies.

## Figures and Tables

**Figure 1 jcm-09-01973-f001:**
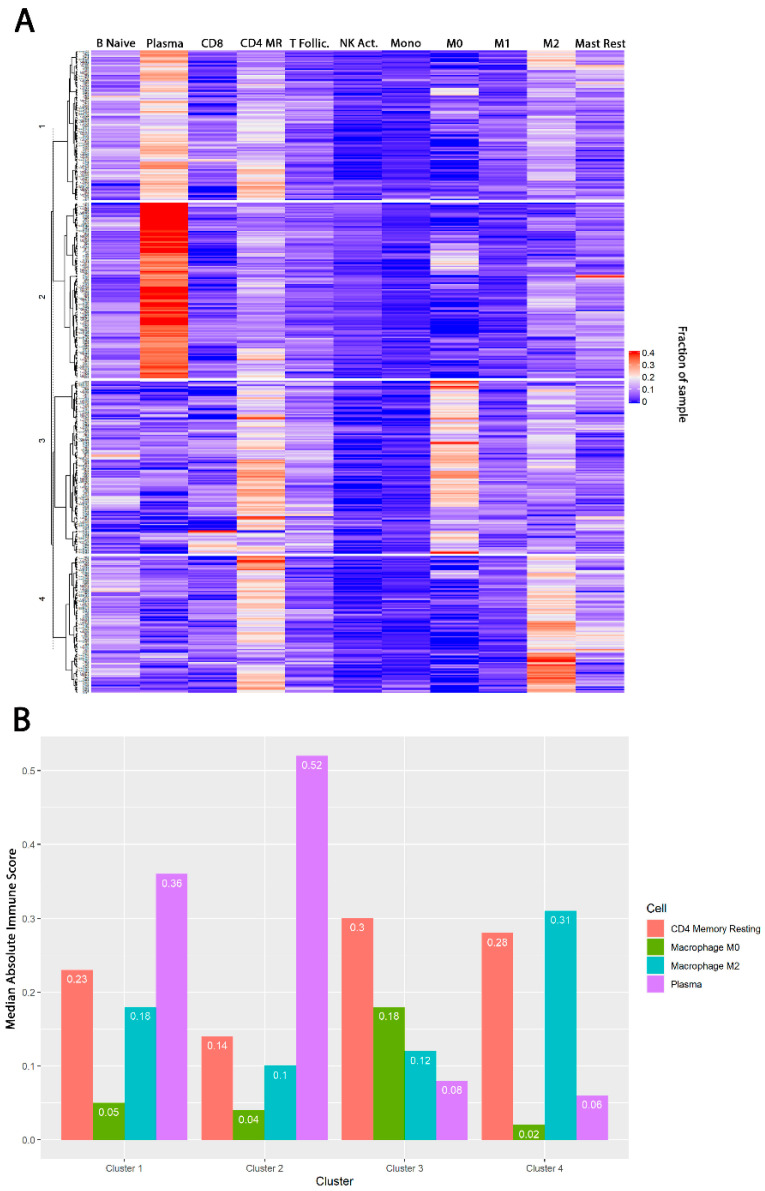
Panel (**A**) displays the clusters of immune populations in the PRAD (prostate adenocarcinoma) TCGA dataset. The columns display the individual immune cells represented in each column, while the rows display the unique identifiers within each cluster. CD4 MR represents CD4 Memory Resting T cells. Cluster 1 is CD4 MR^High^Plasma Cell^High^M0^Low^M2^Mid^. Cluster 2 is CD4 MR^Low^Plasma Cell^High^M0^Low^M2^Low^. Cluster 3 is CD4 MR^High^Plasma Cell^Low^M0^High^M2^Low^. Cluster 4 is CD4 MR^High^Plasma Cell^Low^M0^Low^M2^High^. Panel (**B**) displays the median cell infiltration, derived from CIBERSORT absolute immune fraction, for the most prominent cell types in the TCGA PRAD database by cluster. Abbreviations: T follic, T follicular helper T cells; NK Act, activated natural killer cells; Mast Rest, resting mast cells.

**Figure 2 jcm-09-01973-f002:**
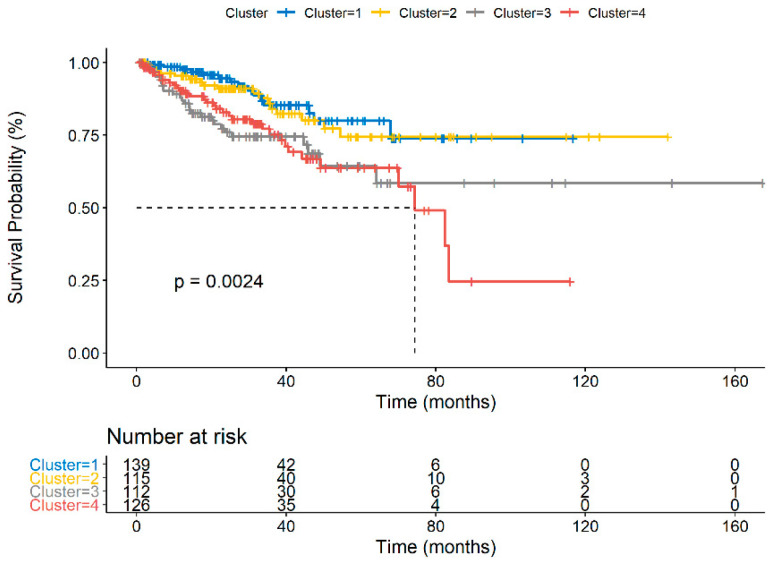
Figure 2 demonstrates the Kaplan–Meier curve for progression-free survival for each of the clusters in the TCGA PRAD cohort. Clusters are color coded, with a legend at the top of the figure. Number at risk at various time points is displayed at the bottom of the figure. Pairwise log-rank comparisons were conducted for each of the curves listed, and results displayed within the figure.

**Figure 3 jcm-09-01973-f003:**
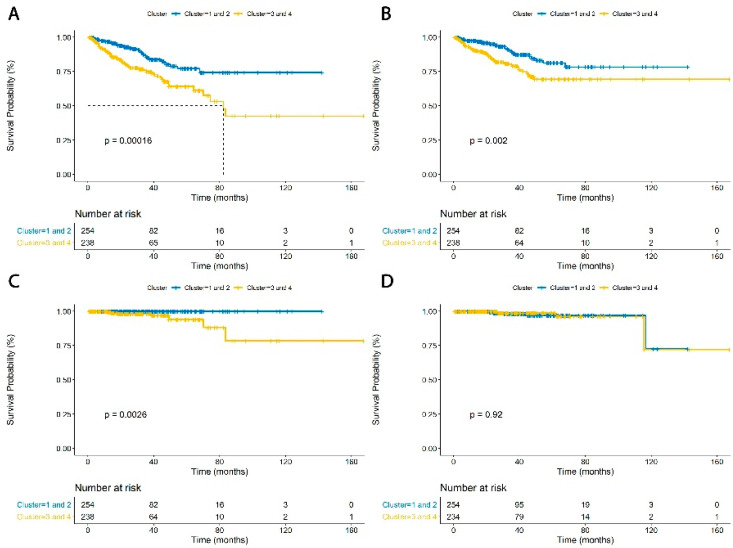
Panel (**A**) demonstrates the Kaplan–Meier curve for progression-free survival for grouped clusters 1 and 2 versus grouped clusters 3 and 4 (HR 2.243, 95% CI 1.458–3.450, *p* = 0.0002). (**B**) Biochemical recurrence (HR 2.128, 95% CI 1.302–3.479, *p* = 0.0026). (**C**) Distant metastasis (all events occurred in clusters 3 and 4, *p* < 0.0001). (**D**) Overall survival (HR 0.9319, 95% CI 0.2488–3.49, *p* = 0.917).

**Figure 4 jcm-09-01973-f004:**
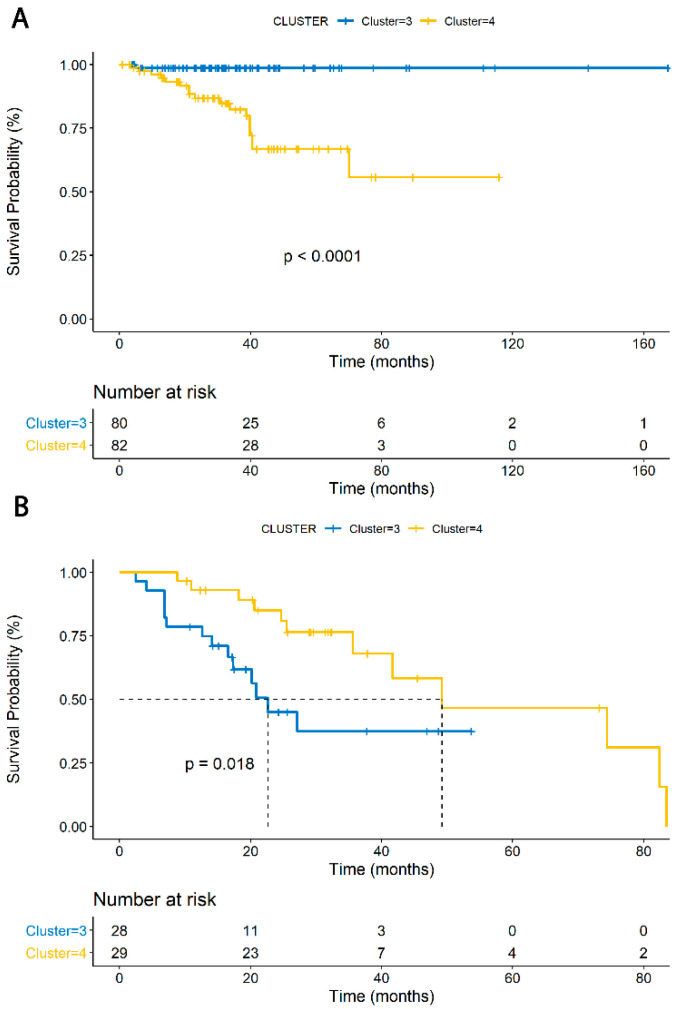
Panel (**A**) demonstrates progression-free survival for only patients receiving surgery in cluster 3 vs. 4, with cluster 4 displaying significantly worse progression-free survival (HR 18.1, 95% CI 2.41–135.5, *p* < 0.0001). Panel (**B**) displays progression-free survival for patients receiving both surgery and radiation in cluster 3 vs. 4, with cluster 3 displaying significantly worse progression-free survival (log-rank *p* = 0.018; HR 0.368, 95% CI 0.1568–0.8642, *p* = 0.0217). The Kaplan–Meier curves are color-coded, with figure legends at the top of each panel.

**Figure 5 jcm-09-01973-f005:**
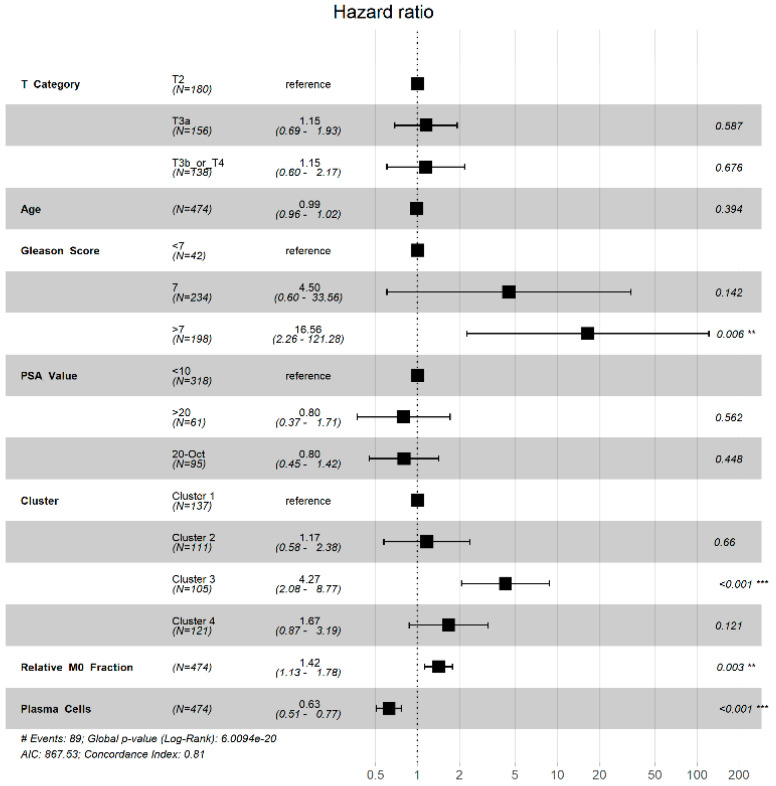
Multivariable analysis with hazard ratio (HR) represented in a Forest plot adjusting for age, pathologic T-stage, Gleason score, baseline PSA value, cluster, and fractions of predominant cells, including M0 macrophages and plasma cells. Clusters, cell types, and clinical risk factors are located in the left column, while hazard ratio calculations are located in the right column. ‘**’ denotes significance, 0.001 > *p* > 0.05, and ‘***’ denotes significance, *p* < 0.001.

**Table 1 jcm-09-01973-t001:** Demographic and treatment data for all patients in the TCGA PRAD cohort, divided by cluster.

	Cluster 1 (*n* = 139)	Cluster 2 (*n* = 115)	Cluster 3 (*n* = 112)	Cluster 4 (*n* = 126)
Median Age (years)	59.9	61.1	62.5	64.7
Median PSA (IQR)	7.2 (5.0–11.6)	7.6 (5.6–9.9)	7.5 (4.8–12.1)	7.5 (5.0–11.3)
Gleason Score				
≤6	20 (14.4%)	10 (8.7%)	6 (5.4%)	8 (6.3%)
7	84 (60.4%)	72 (62.6%)	46 (41.4%)	41 (32.5%)
8-10	35 (25.2%)	33 (28.7%)	58 (51.8%)	75 (59.5%)
Pathologic T category				
pT2	53 (38.1%)	46 (40.0%)	38 (33.9%)	46 (36.5%)
pT3a	53 (38.1%)	36 (31.3%)	31 (27.7%)	38 (30.2%)
pT3b	28 (20.1%)	32 (27.8%)	35 (31.3%)	35 (27.8%)
pT4	4 (2.9%)	0	4 (3.6%)	2 (1.6%)
Risk Group				
Low	14 (10.1%)	5 (4.3%)	3 (2.7%)	7 (5.6%)
Intermediate	74 (53.2%)	66 (57.4%)	42 (37.5%)	38 (30.2%)
High	42 (30.2%)	39 (33.9%)	51 (45.5%)	60 (47.6%)
Zone of Origin				
Peripheral	26 (18.7%)	37 (32.2%)	30 (26.8%)	42 (33.3%)
Central	1 (0.7%)	0	1 (0.9%)	2 (1.6%)
Transition	2 (1.4%)	2 (1.7%)	3 (2.7%)	1 (0.8%)
Multiple	32 (23.0%)	22 (19.1%)	32 (28.6%)	41 (32.5%)
TMPRSS2:ERG Fusion				
Wild-Type	91 (65.5%)	66 (57.4%)	71 (63.4%)	84 (66.7%)
Fusion	47 (33.8%)	48 (41.7%)	40 (35.7%)	42 (33.3%)
Median Follow Up (months)	28	34.3	31.2	30.8
Recurrence				
BCR	10 (7.2%)	15 (13.0%)	22 (19.6%)	21 (16.7%)
LRR	3 (2.2%)	0	1 (0.9%)	3 (2.4%)
Metastasis	0	0	2 (1.8%)	5 (4.0%)
Death	2 (1.4%)	3 (2.6%)	3 (2.7%)	1 (0.8%)
Radiation				
Adjuvant	11 (7.9%)	12 (10.4%)	19 (17.0%)	21 (16.7%)
Salvage	7 (5.0%)	5 (4.3%)	9 (8.0%)	8 (6.3%)
Median Dose (Gy)	66.6	67.5	70	66
Median Fractions	33.5	35	35	34
Salvage Systemic Therapy	13 (9.3%)	16 (13.9%)	25 (22.3%)	25 (19.9%)

Abbreviations: PSA, prostate-specific antigen; IQR, interquartile range; BCR, biochemical recurrence; LRR, locoregional recurrence.

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
