# Peer review of "Tumor Immune Microenvironment Clusters in Localized Prostate Adenocarcinoma: Prognostic Impact of Macrophage Enriched/Plasma Cell Non-Enriched Subtypes"

_jcm, 2020, doi:10.3390/jcm9061973_

Round 1

Reviewer 1 Report

Peer review. Tumor immune microenvironment clusters in localized prostate adenocarcinoma: prognostic impact of macrophage enriched/plasma cell non-enriched subtypes

The authors present data using a valuable method of translating bulk tissue RNA sequencing data into assumptions of cellular subtypes. The inferred immune cellular subtypes are then subjected to cluster analysis that identifies 4 clusters of patients that display somewhat distinct immune cell population patterns. I have issues with the way that the clusters are compared. Some key points are highlighted below.

  1. Figures are low quality resolution. This should be fixed in the final submission.
  2. What is the y-axis on figure 1B. “Value” is not a suitable label for this axis.
  3. Figure 1B. A big issue is the analysis of TCGA PRAD dataset (n = 496) in CIBERSORT. How many patient samples were analyzed in CIBERSORT? If all patients were analyzed, then there should be some statistics associated with the figure 1B values.
  4. In the “macrophage M0 high” cluster # 3, visual analysis of the heatmap suggests that only ~50% of this cluster are actually M0 high. Can the authors comment on this? Do you observe any prognostic significance if this cluster is split into M0 high and M0 low? If not, it may become hard to correlate prognostic significance with M0 macrophage levels in the PRAD dataset.
  5. The fractional difference of 0.13 (between M2 macrophage low cluster 1 and M2 macrophage high cluster 4) is marginal (statistical significance?). Can the authors comment on this and highlight in the manuscript why they suggest this size effect could be significant with regards to prognosis?
  6. Figure 3. The meaning of the Kaplan-Meier curves on metastasis and overall survival when the data is so minimal (7 total metastases, 9 total deaths) is questionable.
  7. Overall, the individual figures/results sections are poorly summarized within the results section. It would be helpful to have a sentence that summarizes the findings of the analysis performed which can then introduce or lead into the next results section to provide an easier document to read.

Author Response

Dear Reviewer and Editors,

Thank you for your insightful commentary. We have addressed all the reviewer comments as below.

Reviewer 1 comments:

  1. Figures are low quality resolution. This should be fixed in the final submission.

Thank you.  This has been corrected.  All figures are at least 1000 pixels width/height and 300 dpi resolution or higher. All images are in TIFF format, except for figure 3 and supplemental Figure 1 due to excessively large size, which are in .PDF format in the .zip file.

  1. What is the y-axis on figure 1B. “Value” is not a suitable label for this axis.

Thank you.  This has been corrected.  Value on the y-axis of Figure 1B has been changed to “Median Absolute Immune Score,” which is the CIBERSORT output for the level of absolute infiltration of each immune cell type.

  1. Figure 1B. A big issue is the analysis of TCGA PRAD dataset (n = 496) in CIBERSORT. How many patient samples were analyzed in CIBERSORT? If all patients were analyzed, then there should be some statistics associated with the figure 1B values.

Thank you.  This has been addressed.  We used the entire available PRAD dataset. A supplemental table has been added to clarify the characteristics of the cohort used, as well as lines 141-147 in the main body of the text.

  1. In the “macrophage M0 high” cluster # 3, visual analysis of the heatmap suggests that only ~50% of this cluster are actually M0 high. Can the authors comment on this? Do you observe any prognostic significance if this cluster is split into M0 high and M0 low? If not, it may become hard to correlate prognostic significance with M0 macrophage levels in the PRAD dataset.

The reviewer brings up an excellent points.  We do agree that the clusters are heterogeneous, however less heterogeneous than the overall cohort of all-comers. Given the increased median absolute cell infiltration on the bar plot, and the nearly universal significance on pairwise comparisons, we believe the unsupervised hierarchical clustering model worked to cluster the tumor samples in a manner demonstrating the least feasible intra-cluster heterogeneity and greatest feasible inter-cluster heterogeneity. To further support our results, we have also now included M0 macrophage cell fraction and plasma cell fraction as continuous variables in the multivariable analysis (Updated figure 5, line 245, legend updated in lines 248-249), which was initially not included, in an attempt to further demonstrate the significance of these cell types. The described changes to the results can be found in lines 234-241 in the text.

  1. The fractional difference of 0.13 (between M2 macrophage low cluster 1 and M2 macrophage high cluster 4) is marginal (statistical significance?). Can the authors comment on this and highlight in the manuscript why they suggest this size effect could be significant with regards to prognosis?

Reviewer comments addressed.  Although it may seem as though the difference is marginal upon first glance, the M2 content of cluster 4 is in fact a 72% increase over the content of cluster 1. Additionally, we have validated that the M2 content of cluster 4 is significantly different from cluster 1 (supplemental Table 1, lines 141-147).  Overall, it is likely a multifactorial effect from multiple immune subsets contributing to poor prognosis, and we feel this is displayed best through clustering methodology. For example, cluster 4 also demonstrated a significant decrease in plasma cells compared to cluster 1. We have emphasized this further on lines 31, 34 and 232. In the discussion, we have further expanded on and emphasized the potential role of plasma cells in lines 270-280, 294, and 300.

  1. Figure 3. The meaning of the Kaplan-Meier curves on metastasis and overall survival when the data is so minimal (7 total metastases, 9 total deaths) is questionable.

We completely agree that this is a limitation of TCGA as mentioned in the limitations section.  We have now specifically addressed these points in the discussion (lines 306-308), and we recognize that this is a significant limitation of the TCGA PRAD dataset.

  1. Overall, the individual figures/results sections are poorly summarized within the results section. It would be helpful to have a sentence that summarizes the findings of the analysis performed which can then introduce or lead into the next results section to provide an easier document to read.

Thank you.  Reviewer comment addressed.  At the end of each results section, there is now a line summarizing the results, before moving to the next paragraph (Lines 203-204, 216-217, 219-220, 241-243).

We thank you again for your comments, and for considering our manuscript for publication in Journal of Clinical Medicine. We look forward to hearing from you.

Reviewer 2 Report

In this manuscript, the authors utilize the CIBERSORTX tool to characterize and cluster certain immune populations infiltrating the tumor tissue in prostate cancer patients to understand the correlation to prognosis. Following are my concerns.

  1. The major concern with the study is that the authors present data to show tumor associated macrophages correlate with poor prognosis. This is well-known, not just for prostate cancer but for several other cancer types. The study does not add any new knowledge.
  2. Did the authors look at other relevant immune population- Tregs, CD8+ T cells, M1 macrophages, etc.
  3. The data shows macrophages to be the sole determinant of prognosis. I am not sure about the relevance of the levels of other cell types in the cluster.
  4. The authors have to substantiate the study with immunohistochemistry/ flow cytometry analyses.
  5. What is the cut-off for the levels of each cell type and what is the criteria? From figure 1B, M2 macrophage levels in cluster 1 looks intermediate, not low.
  6. In figure 4, the authors analyze PFS to show that among patients receiving surgery alone, cluster 4 abundance leads to poor prognosis. The authors don’t analyze or discuss what that observation means.

Author Response

Dear Reviewer and Editors,

Thank you for your insightful commentary. We have addressed all the reviewer comments as below.

Reviewer 2 comments:

  1. The major concern with the study is that the authors present data to show tumor associated macrophages correlate with poor prognosis. This is well-known, not just for prostate cancer but for several other cancer types. The study does not add any new knowledge.

Thank you.  Reviewer comment addressed.  It has been shown previously that tumor-associated macrophages are correlated with poor prognosis (as acknowledged in lines 250-261 of the manuscript), although these were previously largely thought to only represent M2-macrophages. The novelty of this analysis lies in the discovery of M0-macrophages (enriched in cluster 3) as a prognostic factor, especially after receiving radiation, as the role of tumor-infiltrating undifferentiated macrophages has less data in the literature, especially in prostate cancer, to date. We have added a line in the conclusion, line 288, to attempt to emphasize this point.  Additionally, others have attempted to investigate individual cell types, but our methodology was to interrogate immune cell type clusters, accepting that there may be unknown interactions of different immune phenotypes.

To expand on this point for the benefit of the reviewer, M0 macrophages are indeed their own unique subset of macrophages, with expression markers and functionality that are distinct to this subpopulation. Of the 547 genes in the LM22 signature genes file used for immune cell subset deconvolution, there are 33 genes that are significantly differentially expressed by M0 macrophages, and 25 of these are expressed solely by M0 macrophages and not M1 and M2 subsets. Using the MSigDB cancer hallmarks gene sets to characterize these genes, the most common hallmarks characteristic of M0 macrophages are inflammatory response (C5AR1, CCL22, CCL7, CSF1, MARCO), IL6-JAK-STAT3 signaling (CCL7, CSF1, CXCL3), NF-KB-TNF-A signaling (CSF1, CXCL3), and KRAS signaling (MMP9, PPBP).

  1. Did the authors look at other relevant immune population- Tregs, CD8+ T cells, M1 macrophages, etc.

The reviewer brings up an excellent question.  Other immune populations were explored, and were not significantly correlated with PFS. The relative fractions of each of these immune cell types can be visualized in Supplemental Figure 1. What was displayed in the main body of the text was solely those immune populations that were significantly enriched in the TCGA dataset (non-zero median infiltration values). For example, cluster 4 also demonstrated a significant decrease in plasma cells compared to cluster 1. We have emphasized this further on lines 31 and 232. In the discussion, we have further expanded on and emphasized the potential role of plasma cells in lines 270-280, 294, and 300. 

  1. The data shows macrophages to be the sole determinant of prognosis. I am not sure about the relevance of the levels of other cell types in the cluster.

Thank you.  Reviewer comment addressed.  Macrophages (and in particular M0 macrophages) and plasma cells were the focus of this analysis and discussion, because of their dominant presence (or notable absence) in the poorly prognostic tumor samples. Additionally, these cells were found to be significantly correlated with PFS on multivariable analysis. We initially chose not to include the individual cells in multivariable analysis for the reason you have mentioned - it is likely a multifactorial effect from multiple immune subsets that contributes to poor prognosis, and this is displayed best through clustering methodology. However, we have now included this with M0 macrophages and plasma cells to address reviewer comments.

  1. The authors have to substantiate the study with immunohistochemistry/ flow cytometry analyses.

Thank you.  We respectfully disagree and the reviewer knows this is not possible in the TCGA data.  The novelty and utility of CIBERSORT lies in its ability to bypass the need for immunohistochemistry and flow cytometric analysis in a basic science laboratory, instead performing a sort of “in silico flow cytometry.” (https://www.ncbi.nlm.nih.gov/pmc/articles/PMC5895181/). It has been validated against traditional flow cytometric and immunohistochemical methodologies and was found to be non-inferior. It has since been utilized in many landmark scientific papers since its creation.

  1. What is the cut-off for the levels of each cell type and what is the criteria? From figure 1B, M2 macrophage levels in cluster 1 looks intermediate, not low.

This is an excellent point, and an oversight on our part. Our cut-off values were originally based on a mean-centered distribution of the whole TCGA PRAD population, but we now see that this does not adequately capture the variety in M2 presence. M2 macrophages in cluster 1 are now classified as “mid.” This has been corrected on lines 27, 139, and 155.

  1. In figure 4, the authors analyze PFS to show that among patients receiving surgery alone, cluster 4 abundance leads to poor prognosis. The authors don’t analyze or discuss what that observation means.

Thank you.  Reviewer comment addressed. This analysis was displayed to demonstrate that although M2-enrichment (cluster 4) correlates with poor prognosis when patients do not receive radiation therapy, in line with previously reported literature, M0-enrichment (cluster 3) carries a significantly poorer prognosis when these tumors are exposed to radiotherapy, indicating a potential immunosuppressive (or tumor-promoting) plasticity that may be induced by radiation on these tumor-infiltrating undifferentiated macrophages. We have updated the discussion on lines 295-299 to further expand on this observation.

We thank you again for your comments, and for considering our manuscript for publication in Journal of Clinical Medicine. We look forward to hearing from you.

Round 2

Reviewer 2 Report

The authors have addressed my concerns and updated the manuscript wherever appropriate.